# Brief communication: A roadmap towards credible projections of ice sheet contribution to sea-level

Andy Aschwanden[1,*], Timothy C. Bartholomaus[2,*], Douglas J. Brinkerhoff[3,*], and Martin Truffer[1,*]

[1]Geophysical Institute, University of Alaska Fairbanks. Fairbanks, Alaska. USA
[2]Department of Earth and Spatial Sciences, University of Idaho. Moscow, Idaho. USA
[3]Department of Computer Science, University of Montana, Missoula, MT, USA
[*]These authors contributed equally to this work.

**Correspondence:** Andy Aschwanden (aaschwanden@alaska.edu)

**Abstract.**

Accurately projecting mass loss from ice sheets is of critical societal importance. However, despite recent improvements in ice sheet models, our analysis of a recent effort to project ice sheet contribution to future sea-level suggests that few models reproduce historical mass loss accurately, and that they appear much too confident in the spread of predicted outcomes. The inability of models to reproduce historical observations raises concerns about the models' skill at projecting mass loss. Here we suggest that uncertainties in the future sea level contribution from Greenland and Antarctica may well be significantly higher than reported in that study. We propose a roadmap to enable a more realistic accounting of uncertainties associated with such forecasts, and a formal process by which observations of mass change should be used to refine projections of mass change. Finally, we note that tremendous government investment and planning affecting tens to hundreds of millions of people is founded on the work of just a few tens of scientists. To achieve the goal of credible projections of ice sheet contribution to sea-level, we strongly believe that investment in research must be commensurate with the scale of the challenge.

## 1 Sea level rise predictions from ice sheet loss

Global sea level rose during the 20th century more than 3 times faster than at any time during the last 2,000 years (Kopp et al., 2016). Over the last several decades, mass loss from the Greenland Ice Sheet has been the fastest growing contributor to this rise (Chen et al., 2017; Rietbroek et al., 2016), currently tracking the upper-end estimates of the Intergovernmental Panel on Climate Change's (IPCC) fifth assessment report (AR5; IPCC, 2013). Sea level rise driven by global warming is expected to continue over the coming century, potentially flooding 14–322 million people per year in 2100, and reducing annual global gross domestic production by as much as 9% (Hinkel et al., 2014). To guide planning for and mitigation of anticipated damages, IPCC published a suite of sea level rise projections for the remainder of the 21st century: its sixth assessment report (AR6; Masson-Delmotte et al., 2021). Effective planning for coming sea level rise necessitates that these estimates be credible, but also that they be accompanied by a defensible assessment of uncertainty (Moon et al., 2020).

Ice sheet models have emerged as the de facto standard for generating estimates of ice sheet contribution to sea level rise and are the basis for AR6's estimates for the next century, particularly when coupled to models that simulate relevant forcing from

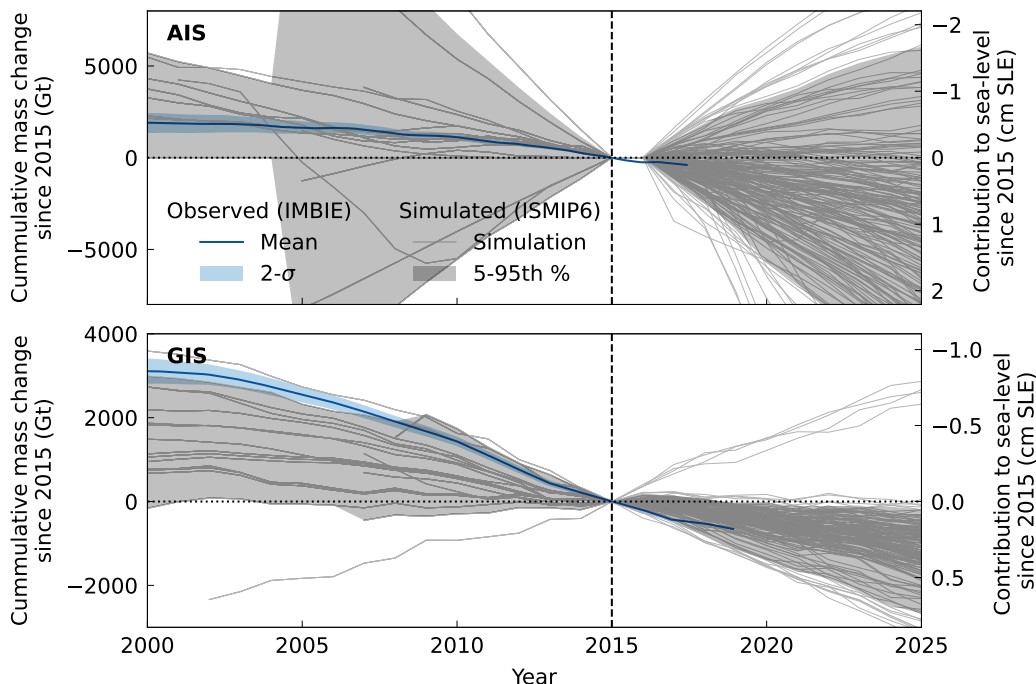

**Figure 1.** Observed and simulated historical mass changes from the Antarctic (AIS) and Greenland (GIS) Ice Sheet 2000–2020 in gigatons (Gt) and centimeters of sea level equivalent (cm SLE). A consensus estimate of observed mass changes (The IMBIE Team, 2019) is plotted in blue along with their respective uncertainties (shaded). The ensembles of ISMIP6 (Goelzer et al., 2020; Seroussi et al., 2020) historical simulations and projections are plotted with dark gray lines, and the 5th to 95th percentile mass loss rates are shown as a 90% credibility interval with light gray shading. Due to the large variance in ISMIP6 historical simulations for Antarctica, the uncertainties of IMBIE are not visible in the plot.

the atmosphere and ocean. However, defensible estimates of the uncertainty in estimates produced by these models remain one of the most challenging goals of scientific inquiry. Here we take ice sheet models to comprise computer code used to predict change in ice sheet mass, including ice dynamics, models of surface mass balance, and models of the ice-ocean boundary.

AR6's estimate of mass change for both Greenland and Antarctica over the next century are based primarily on the Ice Sheet Model Intercomparison Project for CMIP6 (ISMIP6) (Nowicki et al., 2016, 2020; Payne et al., 2021; Seroussi et al., 2020; Goelzer et al., 2020). Through generous collaboration and leadership, 21 groups from around the world contributed 37 different models of Greenland and Antarctic Ice Sheet change through a set of core and optional experiments, and corresponding historical simulations. These simulations had been performed before the latest socio-economic scenarios and climate models were available. Edwards et al. (2021) statistically reproduced the response of these models to climate change and then estimated glacier and ice sheet response to the most recent scenarios and climate model outputs. These Edwards et al. (2021) projections

comprise the AR6 estimate for Greenland. For Antarctica, the AR6 estimate is created by averaging an alternative model

intercomparison (Levermann et al., 2020) with these emulated ISMIP6 projections.

We believe that the results of ISMIP6 represent the state of the art in terms of understanding ice sheet model variability and the breadth of behavior that ice sheet models encompass, and the lessons that it provides will be far reaching both on their own and also with respect to planning additional collaborative efforts to assess uncertainty in sea level prediction. Nonetheless, it is our perspective that the IPCC AR6 estimates of sea level contribution from ice sheets that are based on ISMIP6 may not

be accurate, and that the accompanying uncertainties do not reflect the true breadth of uncertainties associated with ice sheet change. Our skepticism is based on the premise that accurate predictions of the cryosphere's contribution to sea level require that models:

1. Fully characterize uncertainties in model structure, parameters, initial conditions, and boundary conditions.

2. Yield simulations that fit observations within observational uncertainty.

If the first point is not satisfied, then predictive uncertainties are likely to be underestimated. If the second condition is not satisfied, then the distribution of model predictions are likely to be biased relative to reality. For the purpose of credibly projecting mass change, we assert that the models must accurately reproduce observed mass change. Although such validation is insufficient on its own to instill confidence in projections, it is a necessary condition for projection. These points are not just of academic importance, they can lead to a false sense of security when planning coastal infrastructure and preparing for future

sea-level rise, with potentially dire consequences.

Our concern is illustrated by comparing the ISMIP6 simulations of mass loss from the Greenland and Antarctic Ice Sheets between 2000 and 2025 (Goelzer et al., 2020) with observations of mass loss (The IMBIE Team, 2019) (see Methods for details). This 25 year period begins with 15 years of ISMIP6 historical simulations, during which modeling groups were free to select climate forcings necessary to bring their modeled ice sheets to the start of the projection period in 2015. To visualize

how the historical simulations impact the projections, we also show the first ten years of the projection period, during which surface mass balance and temperature anomalies were imposed uniformly on all ice sheet models (Nowicki et al., 2020).

For Greenland, a clear picture emerges (Figure 1), where ISMIP6 simulations systematically underestimate recent relative cumulative mass loss when compared to observations. Indeed, the 95th percentile of the ISMIP6 experiments follows the observed record of mass loss, which implies either or both that the model uncertainty is underestimated and the ensemble is a

biased predictor of cumulative mass change. For Antarctica, the picture is substantially different, with the spread in simulations much larger than observational uncertainty. In this case, many simulations that are not consistent with observations yield a predictive variance that is also not reliable, albeit in the direction of too little specificity.

In an effort to assess why these patterns appear and to guide future efforts, we recast the problem of ice sheet simulation in terms of the characterization of joint probability distributions, and assess how our two conditions above relate to this view-

point. We then sketch a path forward for robustly characterizing the potential ice sheet contribution to sea level over the coming century. We note that our commentary is not intended to be a comprehensive review of uncertainty quantification in ice sheet modelling, but rather to serve as a first attempt to define a consistent language around which community efforts can be dis-

cussed. We also note that the commentary should not be seen as a criticism of the ISMIP6 effort, which we regard as important and successful, and in which some of us have actively participated, but rather as pointing towards important work still to be done.

## 2   Quantifying uncertainties

For the practical problem of predicting the ice sheets' contribution to sea level, we find it useful to adopt a probabilistic framework. In that framework, we seek to establish a credibility bound (say 90%), between which sea level contribution will fall with that pre-supposed probability. Such an interval can readily be constructed from a probability density function (PDF) for the cryosphere's contribution to global sea level by computing quantiles, and thus this is the function that experiments aiming to quantify sea level contribution must correctly characterize. We write this predictive distribution as

$$P(\Delta z | \mathcal{F}), \tag{1}$$

where $\Delta z$ is sea level contribution, and $\mathcal{F}$ represents climate forcing scenarios (i.e. greenhouse gas emissions) expressed as, e.g., Representative Concentration Pathways (RCP) or Shared-Socioeconomic Pathways, which should also be characterized by their own PDF. In this short communication we will not address the issue of uncertainty in the specification of a forcing scenario $\mathcal{F}$ (Team et al., 2010) but concentrate instead on the uncertainties arising solely from ice sheet models.

While interpretation of $P(\Delta z | \mathcal{F})$ is straightforward, its accurate construction is a grand scientific challenge. The standard approach involves running computer programs that approximately solve mathematical equations describing our best understanding of ice sheet physics. In the best case, all facets of a physical system are known (including initial and boundary conditions), the equations describing those systems are complete and deterministic, and the mechanism of solution is perfect. In this idealized situation there is no predictive uncertainty in sea level contribution, and $P(\Delta z | \mathcal{F})$ can be characterized with a single model run. In practice, several types of uncertainties complicate the issue and introduce bias and variance in the predictions. In the following, we discuss these different categories of uncertainty as they pertain to the problem of sea level contribution.

## Model uncertainty

The equations used to describe the physical processes in models are invariably an idealization of reality. Indeed, *all* models are subject to some degree of model error; some physical processes are represented incompletely, while others are omitted altogether. For example, the impact of subglacial hydrology on basal motion (e.g., Bueler and van Pelt, 2015) or the effect of ice mélange (Amundson et al., 2010; Joughin et al., 2020) and iceberg calving (Amaral et al., 2020) on terminus position remain poorly or not represented in numerical ice sheet models, leading to potentially large model uncertainty (sometimes called structural uncertainty). Some hypothesized processes that are often not included in ice sheet models may lead to critical dynamic instabilities that could deeply affect model evolution (DeConto et al., 2021; Sadai et al., 2020). On the other hand, the omission of frictional stresses from wind over an ice sheet surface yields a model that is incorrect yet the resulting error

is negligible. Unfortunately, assessing the non-negligible drivers of model inadequacy is a long and arduous process. The different choices that modellers make in this regard leads to an implicitly defined probability distribution $P(\mathcal{M})$, where a particular model $\mathcal{M}$ is a random (although very likely biased) sample from that distribution. Such model error affects the distribution over sea level contribution as

$$P(\Delta z|\mathcal{F}) = \int P(\Delta z|\mathcal{F}, \mathcal{M})P(\mathcal{M})\mathrm{d}\mathcal{M}. \tag{2}$$

Monte Carlo approximation of this integral is exceptionally challenging because drawing a single sample from $P(\mathcal{M})$ requires the development of a new and (ostensibly) independent ice sheet model, an effort which can take years. However, because many ice sheet models have been developed in parallel, it is now possible to approximate it through model inter-comparison (e.g., Bindschadler et al., 2013; Nowicki et al., 2016; Seroussi et al., 2019; Levermann et al., 2020) or through structured expert judgement (Bamber et al., 2019). We note that an implicit assumption of using models for prediction is that the true data generating process should be contained in $P(\mathcal{M})$, which is a questionable assumption indeed.

**Initial state uncertainty**

Decade to century scale forecasts of ice sheet behavior are sensitive to the initial state, similar to numerical weather forecasts (Vaughan and Arthern, 2007; Aschwanden et al., 2013; Aðalgeirsdóttir et al., 2014). Unfortunately, observations alone are insufficient to define an initial state since not all aspects of an ice sheet state are observable to begin with, necessitating the use of data assimilation to combine sparse observational data with models of varying complexity.

Similar to model uncertainty, initial state uncertainty $\mathcal{I}$ affects the distribution over sea level contribution as

$$P(\Delta z|\mathcal{F}) = \int P(\Delta z|\mathcal{F}, \mathcal{I})P(\mathcal{I})\mathrm{d}\mathcal{I}, \tag{3}$$

where $\mathcal{I}$ is an initial state.

Details vary from model to model, but generally include initial conditions for the conservation of mass (ice thickness and extent), momentum (basal stress distribution), and energy (temperature or enthalpy).

**Parametric uncertainty**

Due to computational and conceptual constraints, there are limits to the level of detail at which processes can be simulated in ice sheet models predicting sea level contribution. For example, the fast and small-scale fracture processes that occur at a marine ice sheet's calving front are more complex than can be reasonably captured in a large scale model with practical time steps. This gives rise to parameters $\mathbf{k} = \{k_1, \ldots, k_N\}$, where $N$ is the number of parameters, specific to a given model which may include different parameterizations than others. These parameters are explicit numerical values that act as the bridge between un-simulated small scale processes and their integrated effects at a practical computational scale. Unfortunately, accurate numerical values for such parameters do generally not exist. This lack of knowledge induces *parametric* uncertainty, for example, different values of thermal conductivity within firn might lead to different predictions of sea level contribution.

The predictive distribution under parametric uncertainty is

$$P(\Delta z | \mathcal{F}) = \int P(\Delta z | \mathcal{F}, \mathbf{k}) P(\mathbf{k}) \mathrm{d}\mathbf{k}, \tag{4}$$

where $P(\mathbf{k})$ is the probability distribution over a given model's parameter values, which we assume to be independent of scenario. Multiple works have approximately evaluated the equation above for either Greenland or Antarctica using Monte Carlo simulation, which is computationally challenging but conceptually simple (NIAS et al., 2016; Schlegel et al., 2018; Aschwanden et al., 2019; Hill et al., 2021): sample a large number of parameter values from $P(\mathbf{k})$, and compute sea level contribution for each sample.

**Aleatoric uncertainty**

Ice sheet models additionally have *aleatoric* uncertainty, i.e. they are subject to irreducibly random processes, most notably the chaotic dynamics present in both atmospheric and oceanic forcings. The predictive distribution under this kind of uncertainty can be decomposed as

$$P(\Delta z | \mathcal{F}) = \int P(\Delta z | f) P(f | \mathcal{F}) \mathrm{d}f, \tag{5}$$

where $f$ represents a specific realization of a random forcing, and $P(f|\mathcal{F})$ is its probability distribution under scenario $\mathcal{F}$. Due to the relatively slow response time of the cryosphere to such forcings, aleatoric uncertainty often contributes little variance to predictions in sea level contribution over practical time scales of decades to centuries. However, in circumstances where these forcings may interact with a critical glaciological instability like the Marine Ice Sheet Instability (Mercer, 1978), aleatoric uncertainty has the tendency of producing 'fat tails', effectively biasing ice sheet evolution towards more extreme mass loss scenarios (Robel et al., 2019). While only a few studies have characterized the distribution over ice sheet responses to aleatoric uncertainty (e.g. Hoffman et al., 2019), and its influence is not precisely known, Monte Carlo simulation can be used to understand the effects of this kind of uncertainty when multiple realizations of forcings are available.

## 3 Assessing the ISMIP6 ensemble through the probabilistic lens

The response of an ice sheet to a given forcing $\mathcal{F}$ may be estimated with Earth System Models directly. At present, however, Earth System Models with built-in interactive ice sheets remain in their infancy (Vizcaino, 2014) and are not yet able to adequately resolve ice sheet processes such as grounding line migration at the necessary resolution, requiring intermediate steps. A common approach, pursued by Goelzer et al. (2020), involves general circulations models to calculate how the global climate responds to a given forcing $\mathcal{F}$, regional climate models to downscale the global climate response to the ice sheet scale, and process models and parameterizations (e.g., surface energy balance models, calving models or frontal ablation models) to interface with ice sheet models. To make the daunting task of estimating ice sheet response to different forcings a tractable community effort, a certain degree of standardization, streamlining, and simplification was necessary. The ISMIP6 steering committee and its working groups prepared data sets that could be used by individual modeling groups, including but not limited to, preparing oceanic (Slater et al., 2019, 2020) and atmospheric (Nowicki et al., 2020) boundary conditions.

Here we consider uncertainty within the ISMIP6 experimental protocols (Goelzer et al., 2020; Seroussi et al., 2020) through the probabilistic frame work outlined above.

**Incomplete consideration of uncertainty**

ISMIP6 integrates over the model uncertainties, including models of ice sheet dynamics, surface mass balance, and ice front position. It does not integrate over uncertainty in parameters. We note that the ISMIP6 protocol allowed modelers to submit as many model setups as they deemed appropriate, in practice, however, each group contributed 1–3 setups. While it is difficult to gauge the magnitude of the resulting underestimation in predictive variance, Aschwanden et al. (2019) suggest that the parametric uncertainty for the Greenland Ice Sheet (inter-quartile range) at 2100 could be up to 0.3 and 12.9 cm SLE for RCP 2.6 and 8.5, respectively, which is larger than the model uncertainty suggested by the ISMIP6 experiments (0.8 and 3.4 cm SLE, respectively). If one takes Aschwanden et al. (2019)'s distributions over model parameters as representing reasonable *a priori* estimates of uncertainty, then the variance in ISMIP6's predictive distribution may be substantially underestimated. Similarly, aleatoric uncertainty is not considered, which has the potential to underestimate mass loss, particularly when dynamic instabilities are likely to play a large role in ice sheet evolution (Robel et al., 2019). Uncertainty also emerges from model initial conditions. A strength of the ISMIP6 protocol was the independence of different modeling groups to select their model initialization protocol.

Taken together, neglecting these additional uncertainties leads to an underestimation of variability in ensemble projections. As a secondary consideration, when comparing model predictions to observations as in Figure 1, this has the effect of ascribing misfit between modeled predictions to model uncertainty, when one of these alternative sources of uncertainty may just as likely be the culprit.

**A biased sample over models**

The implicit hypothesis made when accounting for model error using an ensemble approach is that each model is an independent sample from $P(\mathcal{M})$, where the mode of $P(\mathcal{M})$ is the true data generating process (i.e. reality). However, the models included in the ensemble are not likely to be independent: they share many critical features like numerical methods, parameterizations, and a joint omission of *potentially important physical processes that have not yet been discovered*. We emphatically note that this is not a methodological criticism: it is a challenge that exists generally in science, with analogous situations arising in climate modelling (Qian et al., 2016). We note also that such biases may also arise from incorrectly specified prior distributions over parameters and forcings. Nonetheless, the challenge remains real, as does its potential effect on the credibility and uncertainty of sea level rise projections. As shown in Figure 1, ensemble predictions of mass loss are biased relative to present observations. While the accurate reproduction of observed mass change was not a goal of Goelzer et al. (2020), the credible projection of future mass change was a stated goal. However, there is no reason to believe that a prediction that is biased now does not remain biased in its future predictions.

## 4 A Path Forward

While we do not consider AR6's use of ISMIP6 (Goelzer et al., 2020; Seroussi et al., 2020) and its downstream analysis (Edwards et al., 2021) appropriate for use as the consensus estimate of the ice sheets' contribution to sea level over the next century, it remains a powerful blueprint for the collaborative efforts that the ice sheet modelling community is able to achieve. Building upon the multi-model ensembling approach of ISMIP6, below we offer suggestions on how to more completely account for uncertainties in intercomparison projects.

**Accounting for all sources of uncertainty**

While modelling efforts have captured aleatoric, parametric, initial state, and model uncertainties independently, an effective projection of the ice sheets' contribution to sea level must incorporate all of these sources simultaneously by approximately computing

$$
\begin{aligned}
P(\Delta z | \mathcal{F}) = \int & P(\Delta z | f, \mathbf{k}, \mathcal{M}) \\
& \times P(f | \mathcal{F}) P(\mathbf{k} | \mathcal{M}) P(\mathcal{I} | \mathcal{M}) P(\mathcal{M}) \\
& \times \mathrm{d}\mathbf{k} \, \mathrm{d}f \, \mathrm{d}\mathcal{I} \, \mathrm{d}\mathcal{M}.
\end{aligned}
\tag{6}
$$

To do this, we envision a multi-model ensemble similar to the effort of ISMIP6, but with each model contributing an ensemble of simulations using random parameter values drawn from consensus estimates of the uncertainties associated with parametrically defined physics (cf. Aschwanden et al., 2019; Bulthuis et al., 2019), and with random realizations of climate and ocean forcings (for example, an explicit ensemble of fields characterizing the probability distribution of surface and ocean temperatures) developed in collaboration with their respective modelling communities (cf. Robel et al., 2019). We anticipate that such an effort will yield a distribution of sea level projections that is much broader, and thus less certain, than that presented in recent sea level rise projections (IPCC, 2019). However, we feel that only through modeling what may be considered "unlikely" projections will our community accurately quantify the *a priori* variance in the projections of numerical ice sheet models.

**Conditioning simulations on observations**

While accounting for all sources of uncertainty produces a prior distribution over model projections that appropriately acknowledges the current limits of our scientific understanding, it does not ameliorate the problem of inherent biases in the sampled forcings, parameters, and models. Scientists can add specificity and value to the projected distribution by taking advantage of additional information, such as the observations illustrated in Figure 1. To address both of these problems simultaneously, we advocate for *conditioning* ensemble predictions on relevant observations (Aschwanden et al., 2013). One way of doing this is through Bayes' theorem (often called Bayesian calibration), which states that

$$
P(\Delta z | \mathcal{F}, \mathcal{O}) = \frac{P(\mathcal{O} | \Delta z, \mathcal{F}) P(\Delta z | \mathcal{F})}{\int P(\mathcal{O} | \Delta z, \mathcal{F}) P(\Delta z | \mathcal{F}) \, \mathrm{d}\Delta z},
\tag{7}
$$

where $\mathcal{O}$ is a set of observations, $P(\mathcal{O}|\Delta z, \mathcal{F})$ is the likelihood that some simulation associated with sea level contribution prediction $\Delta z$ agrees with observations, and $P(\Delta z|\mathcal{F}, \mathcal{O})$ is the *posterior predictive distribution* of sea level, which can be thought of as the prior ensemble (Eq. 6) filtered by relevant data.

All ice sheet models already perform this calibration for certain subsets of available observations, e.g. by calibration of basal traction or other parameters to yield observed surface velocity or ice geometry within observational uncertainty. This is a necessary step for models and in many cases part of model specification. However, for the purposes of projecting ice mass change, we argue that the most salient observations on which to condition the prior distribution are measurements of mass change itself (Aschwanden et al., 2013; Aðalgeirsdóttir et al., 2014). It is worth noting that a few works have already performed this Bayesian calibration on observations similar to the mass change observations of Fig. 1, particularly over ensembles meant to capture parametric uncertainty (Nias et al., 2019; Gilford et al., 2020). These studies should be used as a model for future efforts.

Conditioning on observations also requires carefully accounting for the complicated relationship between the time scales of variability in model physics, forcings, and observational uncertainty; the appropriate time scale over which simulations need to show agreement with observations is not (yet) known. The further back in time, the more spatially and temporally sparse observations become, and the larger their associated uncertainties are. Nonetheless, reliable observations of mass change are now available on the decadal time scale (see Figure 1), reducing the likelihood of mistakenly fitting models to short-term fluctuations in weather and ocean conditions. Fortunately, the record of detailed accurate observations is growing continually, soon spanning a climatology (30 years).

By accounting for the broad range of potential *a priori* uncertainties in model projections, and then ascribing predictive weight only to those models that demonstrate skill at reproducing observations, the path towards realistic distributions of sea level contribution over the next century is within reach. Without a large, but realistic, spread of model outcomes it might well be possible that an insufficient number of models remain after fitting to observations.

**Complementary efforts**

Projections made with numerical "high-fidelity" models are computationally expensive and creating ensemble simulations of sufficient size is limited by the availability of computational resources. Training surrogate models ("emulators") with the output of the high-fidelity models can help better characterize sea-level contribution probability distribution functions.

It is worth noting that recent efforts have used ISMIP6 as a basis for further analysis, in particular by training a surrogate model on the ISMIP6 and GlacierMIP output that effectively acts as an interpolant (Edwards et al., 2021). While this interpolant is an effective tool for querying the predictive distribution of sea-level contribution as a function of time and climate scenario *as quantified by ISMIP6*, it inherits the same challenges as its antecedent, namely a lack of accounting for all uncertainty types and a mechanism for bias correction.

Modern machine learning methods show promise to complement established numerical research tools in Earth system science in general, and ice sheet modeling in particular (Reichstein et al., 2019; Edwards et al., 2019; Brinkerhoff et al., 2021; Gilford et al., 2020; Edwards et al., 2021; Jouvet et al., 2021, e.g., ). If numerical and statistical models are paired carefully

and skillfully with structured expert judgment (Bamber et al., 2019), credible projections of ice sheet contribution to sea-level are within reach.

## 5    Meeting the challenge

The potential economic impact of rising sea level has been estimated at over a trillion US$ (Diaz and Moore, 2017) and major world economies including the US consider investing trillions of US$ to prepare for and avert further climate change (Blumer, 2020). Contrasting these staggering numbers, the current funding for research related to sea level rise remains miserly, although exact numbers are not readily available and vary from country to country. During an ISMIP6 planning effort in September 2018, participating modeling groups were polled as to how many simulations they could execute in support of projecting ice sheet contributions to sea level rise. Several groups, none of whom were receiving funding to support these simulations, estimated that they could run 5–10 simulations scheduled amongst their existing commitments. Most likely as a consequence of inadequate funding, no group submitted more than three distinct model setups despite ISMIP6 encouraged participants to explore parametric uncertainties. Projecting future sea level is an effort too severely under-resourced to meet its mission and yet millions of lives and trillions of dollars depend on an accurate, reliable answer.

In order to assess potential impacts of sea level rise, we urgently need to be able to deliberately quantify and then systematically reduce uncertainties. Ice sheet modeling, like climate modeling before it, developed from efforts to address basic science questions. However, despite major advances in the capabilities of ice sheet models and expanding appreciation for the importance of their projections, the funding model of modest grants to address basic science and accomplish incremental model development along the way is unchanged. Internationally, governments directly support development, maintenance, and operation of the Earth System Models that serve as the foundation for CMIP6 (Eyring et al., 2016), and this financial support has contributed to a suite of models that now convincingly reproduce observed climate variability (Jones et al., 2013). It is time to similarly bring ice sheet modeling, both standalone and embedded in Earth System Models, to an operational level and support it with the funding the problem deserves.

The ambitious characterization of uncertainties and ensemble conditioning we propose requires a massive international and inter-agency effort in both model development and improved observational capabilities. We call for professional support for the largely computational sea level projection effort. These resources, in the form of dedicated developers and high performance computing time, will free up scientists to continue basic science, while the global community receives the applied science (i.e., reliable sea level projections) it needs.

The past two decades have shown that ice sheets react to climate far more rapidly than previously thought (Rignot and Kanagaratnam, 2006; Joughin et al., 2014). The study of glaciers and ice sheets has moved from a fringe scientific exercise to a central question of major global economic significance. In response to COVID-19, 18 billion US$ flowed from the U.S. government to fund vaccine development (Tozzi et al., 2020). Appropriate resourcing is possible. While the emergent threat of sea level rise is less abrupt than that from COVID-19, a similarly serious effort is required to reduce uncertainties in sea-level projections.

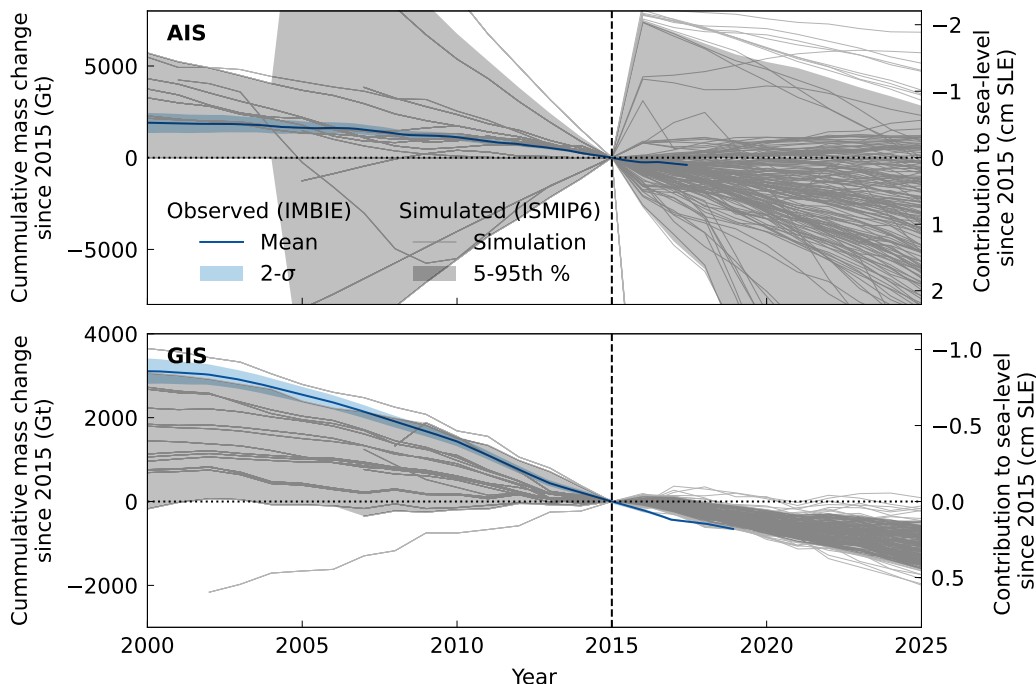

**Figure 2.** Observed and simulated historical mass changes from the Antarctic (AIS) and Greenland (GIS) Ice Sheet 2000–2020 in gigatons (Gt) and centimeters of sea level equivalent (cm SLE). A consensus estimate (The IMBIE Team, 2019) (blue), and their respective uncertainties (shaded). ISMIP6 Goelzer et al. (2020); Seroussi et al. (2020) historical simulations and projection (gray lines) and the 90% credibility interval (light gray shading). Here the control simulation was removed, in alignment with Seroussi et al. (2020); Goelzer et al. (2020)

### Data availability

We downloaded the scalar time-series produced by ISMIP6 for the Greenland Ice Sheet from Zenodo with digital object identifier from https://doi.org/10.5281/zenodo.3939037 (last access: November 2020) (text guidance available from http://www.climate-cryosphere.org/wiki/index.php?title=ISMIP6_Publication_List). The data is split into a historical period (pre-2015) and the projection period (2015–2100). For the projections we used the version where the control simulation was not removed, however a version were the control simulation was removed is shown below (Fig. 2). Removal of the control simulation is intended to account for unforced model drift and mass loss committed as a result of non-equilibrium ice sheet conditions at the start of the simulations. Committed sea level rise is estimated to add an additional 6 mm to simulated sea level rise by 2100 (Price et al., 2011; Goelzer et al., 2020) and thus has little impact on the low bias of recent, simulated mass loss.

For observations, we used a multi-method consensus estimate (The IMBIE Team, 2019).

The analysis was performed using a Jupyter notebook which is available at https://github.com/aaschwanden/ismip6-ipcc.

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
