# Peer review of "Brief communication: A roadmap towards credible projections of ice sheet contribution to sea-level"

_The Cryosphere, 2021_

## Author Comment (AC2)

**Response to reviews:**
**Credible SLR projections**

**General Comment**

During the review process of our manuscript, IPCC AR6 was published. We are delighted that IPCC's discussion of model simulations is now much more nuanced.

We thank all the reviewers for their numerous and insightful comments and their engagement with the paper. We are addressing their comments in detail below. Judging from the comments, it appears that we have not fully succeeded in conveying the purpose of the commentary. Primarily, it was motivated by the sea level rise numbers given by the now published IPCC AR6 and their stated uncertainties. We are concerned that these uncertainties are far too optimistic, and are in fact likely to lead to stated upper limits for sea level rise that are an underestimate, which needs to be known by the larger community (i.e. planners). The second purpose was to outline a way forward for the ice sheet modeling community with, what we think, a useful framework within which to formulate uncertainties and reduce bias.

The commentary may have been perceived by some as a direct attack on the methodology and results of ISMIP6. We wish to stress that this was not the intent. In fact, some of us have actively participated in ISMIP6, and we do not write this from a perspective of a voice that was not heard. The ISMIP6 process was a fantastic and open effort and represents the most mature community ice sheet modeling exercise to date. We sincerely hope that such efforts continue. But we do not think the work of ice sheet modelers is done and the paper is meant to lay out what we think is needed and how we can possibly go about fixing those issues.

We have made revisions in the paper that will hopefully clarify our purpose better. We have also made changes to address many of the detailed comments (see response below).

**Editor Olaf Eisen**

Dear Andy Aschwanden & co-authors,

thank you for submitting your manuscript to TCD, the discussion phase of which will end soon. We received two referee comments and two comments from the community, among the second one is from the ISMIP6 core team. It can be stated that your paper initiated a discussion in the community already before final publication, something which you probably intended.

All comments are very constructive and in some cases particularly detailed, especially those from the ISMIP6 core team, but all in favour of moving forward with this ms though requiring major revisions. "Major" in this case does more apply to the number of revisions/changes you should make rather than to some wrong assumptions. Despite that your manuscript was intially submitted as a brief communication, I ask you to prepare a careful revision and respond to all comments. Some issues are mentioned in various comments (e.g. the role of Antarctica, use of ISMIP6 in the AR6), so you can easily group your responses. Given the interest of the community in this topic I find it more relevant and adequate to clarify all issues in revision rather than to stick to the TC guidelines for "Brief Communications" regarding the number of pages.

With a big thank to the community for providing these comments and the two referees I'm looking forward to your revision.

Best Regards,

Olaf Eisen

**Andrew Shepherd**

A few quick comments:

1/ Slater et al. have compared AR5 projections to IMBIE and they also found that the projections underestimated ice loss. Their conclusion was that the main issue was with SMB and not ice dynamics:

https://www.nature.com/articles/s41558-020-0893-y?proof=t

We added a reference to Slater et al to the introduction.

Our analysis of the ISMIP6 ensemble suggests a more nuanced picture (see Figure below). The ensemble median tracks the IMBIE mean reasonably well for both SMB and ice discharge, suggesting a large variance in individual historical simulations. The difference may be explained by the fact that Slater et al used different data.

[Figure]

2/ Hofer et al. have shown that there is a significant difference between CMIP5 and CMIP6 temperature forcing, particularly in the Arctic, and that this leads to a signficant difference in modelled ice loss for Greenland. Their Fig. 3b shows a cumulative difference between Greenland ice loss under CMIP5 & CMIP6 of something like 1000-2000 Gt by 2020 for RCP8.5. Although this is an extreme scenario, their Fig 5 shows the difference is similar for other pathways. Hofer has suggested that ISMIP6 models may have been forced with CMIP5; as the difference is comparable to the bias shown in your Fig. 1 its probably worth checking this in detail.

https://www.nature.com/articles/s41467-020-20011-8

ISMIP6 planned on using CMIP6 forcing, however the delayed availability of CMIP6 models forced ISMIP6 to use a combination of CMIP5 and CMIP6 models. Barthel et al (2020) describes how ISMIP6 chose a representative selection of CMIP5 models. CMIP6 models indeed seem to show more warming than CMIP5 models, however, Brunner et al (2020) suggest that warming is reduced when CMIP6 models are weighted by their skill in reproducing observations.

Barthel, A. *et al.* (2020) 'CMIP5 model selection for ISMIP6 ice sheet model forcing: Greenland and Antarctica', *Cryosphere*(14(3)pp. 855–879). doi: 10.5194/tc-14-855-2020.
Brunner, L. *et al.* (2020) 'Reduced global warming from CMIP6 projections when weighting models by performance and independence', *Earth System Dynamics Discussions*(pp. 1–23). doi: 10.5194/esd-2020-23.

3/ In your Fig. 1 you show IMBIE as well as 2 individual records of observations which are both included in IMBIE which are both at the upper range of ice losses among the ensemble included in IMBIE. For balance it might be a good idea to show the individual records that are at the lower range also. Or alternatively you could just show the IMBIE alone.

We simplified the plot, now showing IMBIE only.

4/ We have updated the IMBIE assessment for AR6 and I am happy to provide those data should you need them

Andy Shepherd

**Alexander Robel**

This brief communication from Aschwanden et al. argues for a new approach to organizing future iterations of the Ice Sheet Model Intercomparison Project (ISMIP). ISMIP is a community-driven effort to collect and compare simulations from ice sheet modeling groups that are capable of simulating ice sheet evolution relevant to sea level rise from the recent past and the near future. The sixth iteration of ISMIP has recently concluded, with a series of papers in The Cryosphere reporting the outcome from this inter-comparison exercise (Goelzer et al. 2020, Seroussi et al. 2020). There are two main thrusts to the suggestions made in this manuscript:

1. ISMIP should make an effort at formal uncertainty quantification using standardized sets of parameter-perturbed ensembles

2. ISMIP should calibrate projections of future ice sheet change by comparison with observations of past ice sheet change.

Overall, I think this is a valuable manuscript with a message that should be considered by the ice sheet modeling community and particularly those who organize and participate in ISMIP (particularly as ISMIP7 begins to ramp up). There are places where it can be improved, and I have provided constructive suggestions in this regard below, organized into more significant conceptual issues and minor textual/technical issues.

We thank the reviewer for his appreciation of the main arguments within our commentary, and its potential impact on the ice sheet modeling and broader communities. We appreciate his suggestions and respond in turn below.

Major points:

1. It seems accurate to say that section 1 and figure 1 are meant to be the "problem statement" of this manuscript, drawing attention to the shortcomings of the projections from the ISMIP6 multi-model ensemble, with a particular focus on simulated cumulative mass loss from Greenland. While I agree with the general sentiment (and I think most ice sheet modelers would as well), I have some conceptual issues with how the argument is made here that prevent me from being 100% convinced:

(a) It is not obvious that a cumulative metric, such as the one that is used in Figure 1 is the correct one to make the point that there is a mismatch between models and observations. In particular, use of cumulative mass loss in the Figure makes it hard for me to assess whether models are consistently (through time) underestimating mass loss, or just at some point, leading to a persistent offset with respect to observations. The fact that the mismatch between models and observations doesn't appear to grow in time would indicate that the mismatch is mass loss rates is not consistent through time. It would perhaps be helpful to also provide a plot (or a second panel to this plot) showing instantaneous loss rates in simulations and observations to determine whether there are any times during the historical period where models are able to reproduce observed loss.

We think that matching both the trend and the specific (relative) mass are important goals. However, we argue that our focus on the cumulative metric is appropriate here. Much discussion has occurred regarding "ice sheet weather" versus "ice sheet climate". Because of great uncertainty in external forcing it is difficult to imagine the case where the particular observed annual mass variations would be well-matched by the models, if for no other reason than these short term processes are either a) not modelled or b) incorrectly modelled by an external climate model. Instead, looking at time-integrated (i.e. cumulative) behavior yields a somewhat smoother signal associated with the longer term (say, decadal) trends of ice sheet climate. We have added language to the introduction making clear that other metrics of assessing model skill are possible, but that this is the one that we choose.
In addition, we would like to point out that from a societal impact point of view it is the cumulative mass loss number that is of relevance, not the rate.

(b) Related to this issue, I think the following sentence (L45-46) is carrying a lot of the rhetorical weight of this section: "Underestimating recent mass loss likely translates into underestimating mass loss at 2100 as well." This is not obvious to me, and I'm not sure you've provided sufficient evidence here to support this statement. Particularly, it assumes that the sensitivity of the modeled ice sheet change to climate forcing will remain similar (or at leas the gap of this sensitivity between models and observations) between the recent past and the next century. Given that we know there are many aspects of ice sheet dynamics (SMB, MISI, etc) that lead to strongly nonlinear and changing sensitivities, this statement seems hard to support without evidence. I think it is fair to say that a model that can't reproduce the past is unlikely to be skillful in predicting the future, but speculating about the direction of this disagreement seems ill-advised, unless you have evidence to support.

We agree that under-estimating present-day mass loss does not necessarily lead to under-estimating future mass loss although this does seem likely given that, despite being nonlinear, mass loss mechanisms

usually do not switch sign. We are now focusing the discussion on the large variance in simulated historical mass loss, and have removed the reference to the sign of the model error now predicting the sign of the model error later.

(c) At various points throughout this section there is switching between referring to Greenland ISMIP projections, and all ISMIP projections. Yet, you have only shown this mismatch of cumulative loss for Greenland. Could you plot the same thing for Antarctica? Would it show the same mismatch? Given the recent manuscript by Slater et al. (2020) showing a better match from the Antarctic projections, my guess is that it would show that observations are tracking the high end of simulated Antarctic loss, but within the range of ISMIP6. Perhaps this does not make the exact point you are trying to get across here, but it would be a more accurate representation of the full ISMIP6 exercise, which included both ice sheets. Otherwise, focusing your discussion here on Greenland and discussing why the same mismatch might not be true in Antarctica would be a more comprehensive assessment of ISMIP6.

We added Antarctic mass loss to Figure 1, and included a discussion of how its relationship to observations differs from that of Greenland.

(d) It doesn't seem fair to compare observations to simulated mass losses where you have removed the unforced control simulation. It is argued in the Data Availability section that this "is intended to account for unforced model drift and mass loss committed as a result of non-equilibrium ice sheet conditions at the start of the simulations". However, the real Greenland ice sheet was probably not at equilibrium in the latter half of the 20th century, thus you are not making a fair comparison between the two, and potentially biasing to less mass loss over the simulation period. I would suggest not to remove the control simulation.

This is an interesting point. In our initial conversations with ISMIP6 authors, they suggested that we use the simulations with control run removed in agreement with the ISMIP6 manuscripts. However, we agree with the reviewer here that it makes more sense to compare uncorrected model simulations to the observations. We have updated the figure to reflect this, and put the control-corrected version in a supplement. We note that the resulting difference is small and does not affect the qualitative conclusions of this commentary.

2. There has been considerable effort in recent years to improve UQ and Bayesian calibration best practices in ice sheet modeling, which hasn't been cited here, particularly: Schlegel et al. 2018, Bulthuis et al. 2018, Nias et al. 2019, Gilford et al. 2020, DeConto et al. 2021 (among others already cited including the study led by the lead author focused on Greenland). My takeaway from surveying this work is that the field is moving in the right direction, but not all groups have adopted the state-of-the-art practices in UQ and BC, with a large part of the reason being a lack of computational and financial/human resources, which are needed to do these sorts of resource-intensive ensembles. My suggestion would be to modify the message (perhaps softening it) to give credit where it is due (these uncited studies), and indicate how these "best practices" can be integrated into our-community wide intercomparison exercises (or perhaps have a completely separate inter-comparison exercise that is more UQ-focused). My hope would be that making these methods part of the standard intercomparison practice would spur a larger fraction of the community to be working on these problems, even outside ISMIP exercises.

Our intent with this commentary was not to present a full review of UQ practices in ice sheet modelling, but rather to illustrate the specific UQ challenges associated with the study that has been adopted as the official IPCC AR6 estimate, as well as to develop a conceptual framework that will hopefully serve as a vocabulary for describing what sources of uncertainty need to be addressed in future ensemble projections. We have added language to the introduction clarifying that intent. That said, we agree with the reviewer that it will be useful to the reader to find examples beyond what we have already presented of UQ applications. We have thus augmented our discussion of the different types of uncertainty with the references suggested, as well as a few more where appropriate.

3. While the mathematical formalism you adopt here is certainly nice and clean for separating and explaining the various types of model uncertainty, it is not clear that all sources of uncertainty can be separated so cleanly in reality. Particularly relevant to the suggested experimental design for ISMIP (of asking modeling groups to run ensemble simulations with a prescribed set of parameters) is the distinction between P(M) and P(k). Not all models have the same parameterizations. Not all models have the same numerical implementations of the same parameterizations. Some models have processes that no other model includes (e.g., MICI calving, temporally-varying subglacial friction). Thus, the structural model prior and the parameter prior are already convolved in many ways. Sometimes this can be controlled for (i.e. turning off the non-standard process), but sometimes it is built directly into the numerics of the model in a way that hinders simple inter-comparison across models. It would seem useful for the authors to suggest some ways that this issue can be addressed by the ISMIP organizers if this suggestion is to be taken up.

Of course we agree with the reviewer, and enforcing consistent parameterization across models is not what we intended to suggest here. We assume that each model has its own parameters, which we now state explicitly in the section on parametric uncertainty. However, we also note that in Eq. 6 (the integral over all sources of uncertainty), we do explicitly condition the parameters k on M, as suggested by the reviewer.

Minor suggestions:

L23: attendant contribution to sea level exercise

Reworded.

L27: ice sheet change

Ice sheet is usually capitalized when used in conjunction with Greenland or Antarctic, no?

L54-55: Confusing sentence

Updated wording to be more succinct

L62: Similar to point #3 above, since climate and ice sheet geometry feed back on each other, are $P(F)$ and $P(M)$ and $P(k)$ separable?

We think that this largely depends on definitions. If one were to view f as fully exogenous, then yes, the separation holds. This is a reasonable interpretation if we assume that the couplings and feedbacks which might modify, say, surface mass balance occur within the notional confines of M. Of course there might be a more useful way to partition these things depending on the task at hand, but we find it convenient to use this (admittedly not the most well-suited for conveying nuance) perspective.

L65-66: This point about deterministic dynamics is somewhat in conflict with your later points about aleatory uncertainty. We would need to have perfect observations of all initial and boundary conditions along with a complete set of equations to have a system with deterministic dynamics.

Indeed. We were trying to convey precisely the reviewer's point, but the sentence is a bit confusing. We have tried to clarify.

L79: is it actually a random sample?

This is a bit of a philosophical question, but we would argue that, yes, it is random, but also very likely biased. We have added this latter point.

L86: its probably worth it to mention InitMIP here

Added, along with some other relevant ones.

L89: why not?

Reasonable question. Because not all ice sheet state variables are practically observable. We have added this.

L98: the fast and small-scale fracture processes

Added the suggested modifier.

L99: captured in a large-scale model with long time steps

Added this clarification.

L107: simulation of Greenland Ice Sheet evolution

We now include more examples and mention that this marginalization has been approximated both for Greenland and Antarctica.

L112: It has been argued that some ice sheet processes are sufficiently complex and chaotic as to be considered part of aleatory uncertainty: calving, subglacial hydrology, etc.

We don't agree with this. These are not chaotic exogenous forcings, but rather internal dynamics that are insufficiently resolved. We suppose where to draw the line between what constitutes the model versus not-the-model is arbitrary, but since we frequently attempt to include coupled dynamics to simulate these things, they would appear to be part of the model.

L121: worth it to cite Hoffman et al. 2019 here too

Yes, good point. We have included this reference.

L144: sentence starting "Depsite..." is confusing

We have reworded this to be more clear.

L146: parametric uncertainty for the contribution of the Greenland Ice Sheet to sea level exercise

We have made this more specific.

L161: joint omission of...

Agreed that this is the better wording. Updated.

L165-166: observations is used twice in this sentence

Fixed the typo.

L167-168: confusing sentence

We have tried to clarify.

L170: should cite Edwards et al. 2021 here

Done.

L173: ...uncertainties in intercomparison projects. [Again, the point being that such approaches have been used in the field, but not in ISMIP.]

We have made the suggested addition.

L177: to sea level must include all types of uncertainty simultaneously...

Done.

L183: with many realizations of random climate...

We added "random realizations of climate forcing"

L189: uncertainty produces a prior distribution...

Changed.

L200: be a bit more clear - is the argument that modeling groups should not do any of their own calibration for simulations submitted to ISMIP (under the proposed plan)? Does inversion for initial conditions count as calibration? Moreover: should all the calibration be conducted centrally (i.e. after the fact by the ISMIP team using standardized tools)?

We should have been more clear that we were only referring to ex post facto calibration of the ensemble against mass change specifically. This is fixed now.

L209-210: I'm not sure multi-decadal observations of ice sheet change are able to rule out the possible effect of internally driven climate variability. 30 years is not a climatology for ice sheet changes which have intrinsic response time scales of many to decades to hundreds of years. Moreover, one of the challenges of the limited observational record is that climate variability during the observational period may introduce a bias during calibration (e.g. calibrating ice sheet models on the changes that have occurred at Sermeq Kujalleq/Jakobshavn over the past 30 years when they have also experienced a large internally driven warming event.)

We agree that the time scales at which ice sheet "weather" turns into "climate" have not yet been established and that some ice sheet processes have time scales much longer than 30 years. Nonetheless, using a 30 year climatology still seems an acceptable context, in particular since observations prior to the satellite era are too sparse and have too large uncertainties to be used as initial conditions and for calibration. Additionally, each model group was free to initialize their models with a historical run of their own choosing, which could, in theory, include more internally driven ice sheet variability. We are, however, unsure what the reviewer means by a "large internally driven warming event" and so are unable to address this point.

L227: DeConto et al. 2021 and Gilford 2020 useful to cite here too

We added Gilford 2020 as this seems the more appropriate addition, and added that this is only a selection.

L230: the framing here is US-centric (particularly for an EGU journal)

That's a fair criticism, but we think the issue goes beyond the US; we tried to point this out in the revised version.

L234-235: is there any way to include an estimate of how much of OPP budget actually went to modeling/SL projects (even just for 2019). This would be an incredibly useful number to have on the published record.

That would be a bit of a pain to parse, but it's going to be very little. NSF has done very little funding of actual modeling work.

L240-241: I'm a bit confused by this. Is the issue that ISMIP is voluntary or that it isn't financially support at an appropriate level? I'm pretty sure the solution isn't to force modeling groups to participate (making it non-voluntary), but rather to incentivize groups to participate by making funds available to support the work and computation that require some of the changes you are suggesting.

This is a good point.  The issue of course is the broad lack of financial support at an appropriate level (we, for the record, do not support enslaving ice sheet modellers, even if for the good of the rest of humankind).

L256: computing resources, will free ups scientists to continue conducting basic science reserch, while the global community benefits from needed advances in applied science (i.e. reliable sea level projections).

**ISMIP6 Steering Committee**

We post this comment as members of the scientific steering committee of the Ice Sheet Model Intercomparison Project for CMIP6 (ISMIP6): Sophie Nowicki, Antony Payne, Eric Larour, Heiko Goelzer, William Lipscomb, Helene Seroussi, Ayako Abe-Ouchi, Andrew Shepherd.

We welcome comments and suggestions for follow-on extensions for the ISMIP6 protocol. We are pleased to see these suggestions from some of the participants of the original ISMIP6 project.

We thank the ISMIP6 community for their attention to our draft manuscript, and the time they have spent with it improving and correcting our text, and clarifying our message.

**General comments**

1. The commentary in this manuscript is focussed on a single multi-model comparison (ISMIP6 Greenland). Many of the suggestions would similarly apply to other initiatives in the ice sheet modelling community (e.g., ISMIP6 Antarctica, ABUMIP, and LARMIP2), outside of it (e.g., GlacierMIP or even CMIP), and equally to many individual projections. While ISMIP6-Greenland can still be used as an example, we would appreciate clarification on why the authors did not undertake a broader approach (for example, using both ice sheets).

We chose to focus on ISMIP6 because of Fig. 9.19 in the draft AR6. The mismatch in slope between the observed, historical mass change and that of the prediction period piqued our initial attention. Additionally, although the points we make are not unique to ISMIP6, Goelzer et al. (2020) is somewhat unique in the directness with which it projects future mass change from Greenland, stating in the abstract, "The results indicate that the Greenland ice sheet will continue to lose mass in both scenarios until 2100, with contributions of $90 \pm 50$ and $32 \pm 17$ mm to sea-level rise for RCP8.5 and RCP2.6, respectively." The strength and clarity of this assertion, and the apparent community backing behind it (including by one of our author team), distinguishes it from other community ice sheet modeling efforts.

For completeness we added ISMIP6 Antarctica. However, a deep review of the current literature is outside of the scope of our manuscript. We do point out in the manuscript that ISMIP6 is the most mature effort of its kind, so there is some logic in concentrating on that. In addition, AR6 predictions for Greenland focus heavily on ISMIP6.

2. The framing of how ISMIP6 results have entered the AR6 report is problematic. The authors state: "This ISMIP6 distribution has since been adopted as the foundation for the IPCC AR6 consensus estimate of sea level contribution from ice sheets." As of now, the AR6 report has not been released, so it is not yet possible to say how the AR6 authors used ISMIP6 results. IPCC guidelines (Mastrandrea et al., 2010) on uncertainty do not support the use of a single line of evidence as the basis for an assessment, and therefore the AR6 consensus estimates will likely draw from many studies.

AR6 was published in the meantime, and we changed our language accordingly.

3. The ISMIP6 protocol and goals are not always accurately represented in this commentary. For example, participants were not limited to a "single" contribution, so anyone could have contributed to ISMIP6 a series of simulations to sample uncertainty in the way the authors describe. See specific comments below.

This is true. We suggest that limited computational resources and financial support by funding agencies may have forced modelers to contribute fewer simulations than they would have deemed appropriate otherwise. We changed wording in the manuscript to state that each group contributed between 1 and 3 setups.

4. The authors generally do not clearly distinguish ice sheet model uncertainty from forcing uncertainty. Their proposed method of partitioning uncertainty relies on an oversimplification of the ISMIP6 protocol as it ignores complexities not present in, for example, Aschwanden et al. (2019).

The manuscript is explicit on focusing entirely on uncertainties introduced from ice sheet modeling and acknowledges the potentially much larger uncertainties from climate forcings (see section 2).

5. Can the authors more clearly link Sect. 5 to the rest of the paper? This section seems to focus on a separate issue that might be relevant to a different audience. Taken in isolation, the ISMIP6 Greenland projections do not provide a strong basis for the commentary.

Section 5 is meant as a 'call to action' for funding agencies, etc. It is not directly related to ISMIP6, but applies to ice sheet modeling in general. We think it is appropriate to end the paper on this note, with the hope that it helps free up resources that would benefit the whole community and society at large through more credible predictions of sea level change..

**Specific comments**

L. 3: "...few models reproduce historical mass loss accurately"

This is an important argument in the manuscript, but it requires a more nuanced treatment. Questions that need to be addressed are: Can and should the ISMIP6 projections be expected to reproduce observed mass changes? Would reproducing mass change over the relatively short observational period be a meaningful quality criterion for centennial time scale projections? We address these points in more detail below.

We argue that if predicting mass change from an ice sheet is the goal (Goelzer et al, 2020), then observed mass change is the most valuable metric by which a model prediction can be judged. There is, of course, the issue of time scales. We are trying to be more explicit about that. Observations are now starting to span several decades, and the lack of fit to changes on that time scale should at least be some cause for concern.

L. 6: "the future sea level contribution from Greenland may well be significantly higher than reported"

Little evidence is presented for this statement in the main text. If the claim is based on the possibly larger uncertainty, it could mean both higher and lower contributions. We suggest reformulation.

This concern was shared by another reviewer, and we have thus changed this sentence. While the concern is valid, it seems unlikely given the clear underestimate of rates during the observational period.

L. 8-11: "Finally, we note that tremendous government investment ..."

We agree on the need for more investment in ice-sheet and sea-level research. However, if the "significantly volunteer effort" refers to ISMIP6, the wording ("is founded on") may exaggerate the role of ISMIP6 results in government planning. See the comments above on multiple lines of evidence in IPCC assessments.

AR6 is very heavily guided by ISMIP6, so this does not seem such an exaggeration.

L. 20: "accurate"

To be "accurate" implies comparison to observation. Projections are of the future, so it is impossible to assess their accuracy now. We suggest a different phrasing.

Changed to "credible".

L. 21: "defensible assessment of uncertainty"

The IPCC has robust guidelines to ensure that uncertainty is indeed assessed (e.g., Mastrandrea et al., 2010). As written, this sentence calls into doubt the IPCC methods of assessing the uncertainty. Please rephrase.

The IPCC has to base their work on the published literature. They may have clear guidelines, but if the literature underestimates uncertainties, then this propagates directly into the assessment report. Case in point is AR5 compared to observations, which are also largely outside the range of most predictions (see Slater, 2020).

L. 23: "Ice Sheet Model Intercomparison for CMIP Phase 6 (ISMIP6)"

The correct name is "Ice Sheet Model Intercomparison Project for CMIP6 (ISMIP6)".

Changed.

L. 24: "(ISMIP6)"

References for ISMIP6 should include Nowicki et al. (2020) and Payne et al. (2021). Nowicki et al. (2020) contains the description of the experimental protocol which is mostly criticised here.

We added references to Nowicki et al. (2016, 2020) and Payne et al (2021).

*Figure 1:*

This figure is presented as a key line of evidence in Sect. 1, but important choices of selecting data, processing output, and combining results have not been motivated and described with sufficient detail. In particular:

* It is unclear how the uncertainty envelope has been derived. The original figure for the historical period (Fig. 4, Goelzer et al., 2020) does not attempt to show uncertainty in the model results, but simply reports the ensemble. Please clarify how the 90% credibility interval was derived.

It's simply the envelope of the [5, 95]-percentiles of the historical simulations and of all members of the projection period ensemble. We changed the notation.

* The uncertainty envelope of, e.g., IMBIE depends on the choice of assuming fully correlated or uncorrelated errors when accumulating uncertainties (compare again Fig. 4 in Goelzer et al., 2020). Can you motivate your choice for the narrower envelope and discuss the implications?

This is a valid point. We show the cumulative uncertainties as provided by the publicly available IMBIE data. In the revised version we now show the 2-sigma level for IMBIE. While adding fully correlated errors would result in more historical simulations being with the IMBIE uncertainties, the bigger picture would not change: that it's the 95th percentile of ISMIP6 that tracks the IMBIE mean, demonstrating the strong tendency of ISMIP6 historical simulations to underestimate observed mass loss.

* Please discuss the conceptual difference between the historical experiments (until 2014) and the projections (2015+). While modellers were free in their simulation of the historical period, the projections were tightly constrained by CMIP model output. Combined analysis across those experiments (e.g., 2008–2020) is therefore difficult to interpret. See also the comment on L. 45 below.

We tried to clarify by adding "Our concern is illustrated by comparing the ISMIP6 simulations of mass loss from the Greenland and Antarctic Ice Sheets between 2000 and 2025 \citep{ISMIP6-Greenland} with observations of mass loss \citep{IMBIE2019} (see Methods for details). This 25 year period begins with 15 years of ISMIP6 historical simulations, during which modeling groups were free to select climate forcings necessary to bring their modeled ice sheets to the start of the projection period in 2015. To visualize how the historical simulations impact the projections, we also show the first ten years of the projection period, during which surface mass balance and temperature anomalies were imposed uniformly on all ice sheet models \cite{Nowicki2020}." .

L. 27-28: "ISMIP6 produced probabilistic distributions of projected sea level contribution"

ISMIP6 did not produce probabilistic results – it presented ensembles with no probabilities attached. Others (e.g., Edwards et al., 2021) have used these ensembles to make probabilistic assessments, but their analysis includes additional information and not simply the ISMIP6 results.

We reworded the paragraph.

L. 31: "This ISMIP6 distribution has since been adopted as the foundation for the IPCC AR6 consensus estimate."

What is the basis for this statement? See the general comment above.

The basis was the draft report. But this is now referenceable, so we have corrected this absence.

L. 35: "Our skepticism regarding the ISMIP6 projections is based on the premise that accurate predictions of the cryosphere's contribution to sea level require that models:

1. Fully characterize uncertainties in model structure, parameters, initial conditions, and boundary conditions.

2. Yield simulations that fit observations within observational uncertainty."

Although the requirements are laudable, it is nearly impossible for any study to achieve both, let alone a large multi-model project such as ISMIP6. To "fully characterize uncertainties" is demanding indeed, but this is not a problem within a single study, as long as other research addresses other uncertainties. It is also impossible for a model to fit all available observations within observational uncertainty, unless the model is overtuned. One can argue that particular observations are supremely important, but it is not obvious that recent mass loss is more important than, say, an accurate simulation of observed ice extent, thickness, and velocity.

The call for a full characterization is demanding indeed, by design. Whether this is presently or ever possible does not matter. What one would hope, as models and forcings get better, that the misfit reduces. The problem with AR6 is that it creates the distinct impression that uncertainty is much lower than it actually is, at least for Greenland. We believe that the main quantity of interest makes a natural candidate for validation but any thorough validation and calibration process should be complemented by other fields as pointed out by Aschwanden, Adalgeirsdottir and Khroulev (2013). Ice thickness would be a poor metric because of the high observational uncertainty.

Additionally, mass loss is the most societally relevant quantity. We have added text in the manuscript stating that it is our opinion that recent mass loss ought to be an important metric if future mass loss estimation is the main goal.

"It is also impossible for a model to fit all available observations within observational uncertainty, unless the model is overtuned."

We disagree with this assertion. For a model to be credibly used for specific quantitative prediction it needs to be validated, which is to say that the mathematical representation of the system and its mechanism of solution needs to be proven suitable for answering the question that is being asked. For ice sheet models (which rely on significant empiricism) being asked to project sea level rise, it is hard to imagine how successfully predicting past sea level rise would not be considered a necessary condition for such a model to be valid. In the event that the model's predictions and observations diverge, the estimate of predictive uncertainty must be commensurately increased.

Although this paragraph is set up to elaborate both requirements, the subsequent analysis of Fig. 1 is based only on the second requirement. See the comments above on the augmentation of ISMIP6 results.

We do not understand this criticism: an entire section of the paper is devoted to characterizing uncertainties.

L. 43: "Most simulations underestimate recent (2008–2020) mass loss."

The period 2008–2020 straddles the ISMIP6 historical period (ending in 2014), during which modellers used the forcing of their choice, and the future period (2015+), when forcing was provided by climate models. Mass loss from 2015 reflects, in part, natural variability that would not be reproduced by the climate models.

Reworded, see reply to "Please discuss the conceptual difference between the historical experiments ".

Why focus on 2008–2020? Figure 1 (and Fig. 4 in Goelzer et al., 2020) start before 2008, so the statement applies also to a longer time period.

See above.

L. 45: "Underestimating recent mass loss likely translates into underestimating mass loss at 2100 as well."

This is not necessarily true. As pointed out above, it is important to distinguish the historical period (before 2015) from the projections. Generally speaking, in order for ice sheet mass loss to be accurately simulated for the recent past (2008–2020), two things are required: The climate forcing should be accurate, and the ice sheet model should accurately represent the processes translating this forcing into mass loss.

Modellers were free to choose their own forcing for the historical period; in most cases, they used SMB output from regional atmosphere models such as RACMO and MAR. Some ice sheet models may have applied SMB forcing that was biased positive for the period 2008–2014. Also, most models did not apply forcing to outlet glaciers before 2015.

ISMIP6 climate forcing from 2015 onward was derived from the CMIP5 and CMIP6 Earth System Model (ESM) ensembles. A known complication of this forcing is that interannual variability (known to be important in determining Greenland's mass budget) is seldom in phase with the observed climate. This significantly complicates a model–observation comparison over a short period of 12 years.

For these reasons, models that underestimated mass loss during 2008–2020 might have been responding realistically to biased forcing. To demonstrate that they underestimate mass loss at 2100, one would need to argue that (1) the ESM-derived SMB forcing through 2100 is biased positive, and/or (2) the models underestimate recent mass loss when forced with an accurate SMB and output glacier forcing.

We have removed the specific claim regarding the sign of the future potential predictive error.

It is certainly possible that the choices made by individual modelers during the historic period were all biased in the direction of too low a surface mass balance, with the ice sheet models themselves faithfully translating this signal in an unbiased way to a mass loss that was, demonstrably, biased relative to historical observations. If this is the case, then upon reaching 2015 and the beginning of the ensemble CMIP5/CMIP6 forcing, if one assumes that forcing to be unbiased, then the ensemble may yet yield unbiased predictions. We do not discount this possibility, however, we also do not find sufficient

evidence to state with high confidence that either the assumption of an unbiased ice sheet model or the assumption of an unbiased projected climate model forcing are good ones. Again, these models are not validated: to be considered such for the purposes of projection, they need to reproduce reality with skill.

L. 61: "In this short communication we will not address the issue of uncertainty in the forcing F (Team et al., 2010) but concentrate on the uncertainties arising solely from ice sheet models."

There appears to be some confusion about this separation. In ISMIP6 (unlike Aschwanden et al., 2019), the SMB and outlet glacier forcing are prescribed and are not part of the ice sheet model formulation. Much of what can be assigned to parametric uncertainty in Aschwanden et al. (2019) has no equivalent as ice sheet model uncertainty in the ISMIP6 projections, but rather is connected to uncertainties in the forcing. Thus, the uncertainty framework described here does not apply in the same way to ISMIP6. Goelzer et al. (2020) state explicitly that we did not sample RCM uncertainty (which could loosely map to some of the uncertainties in PDD factors), but these complexities are not discussed here.

Since uncertainty in the forcing (SMB and outlet glaciers) could account for the issues highlighted by Fig. 1, it seems appropriate to address that uncertainty.

The line between what constitutes an external forcing versus a part of the ice sheet model is, of course, blurry. However, it is clear that at some point, the translation from a climate model to a specific mass balance required the use of a model of the melt process. Whether explicitly calculated as part of the ISMIP6 experiments or not, this is still a part of the process of using a model to make predictions, and we argue that its associated uncertainty needs to be considered as well. Similarly, the prescription of boundary position implies that some sort of e.g. calving parameterization is being used (whether explicitly defined or not), which is also subject to uncertainty. What we hoped to do, is to at least draw the blurry line between ice sheet and climate *somewhere*, since we felt it out of scope to attempt a discussion of the uncertainties associated with climate models. We have added text in the introduction to clarify what we consider to be "the ice sheet model" for the purpose of this commentary, specifically "here taken to comprise computer code used to predict change in ice sheet mass, including ice dynamics, models of surface mass balance, and models of the ice-ocean boundary".

L. 102: "This lack of knowledge induces parametric uncertainty, for example, different values of thermal conductivity within firn might lead to different predictions of sea level contribution."

This is true, but the thermal conductivity within firn is relevant only for the RCMs computing the SMB. So this example pertains to MAR and RACMO, but not the ensemble of ISMs.

Once again, the conceptual boundaries of an ice sheet model are blurry. Here, we consider firn models to be a part of the ice sheet model. Which computer runs the code is less important than whether or not its associated uncertainty is correctly quantified. At any rate, this was only meant as an example, there are of course many other parameters with associated uncertainties.

L. 134: "(Slater et al., 2020)" , "(Barthel et al., 2020)"

Please cite two papers by Slater et al.: Slater et al. (2019) and Slater et al. (2020). The reference to Barthel et al. (2020) should be replaced by Nowicki et al. (2020), since only the latter paper shows atmospheric boundary conditions.

Done.

L. 134-137: "To allow a wide range of modeling groups to participate, concessions had to be made, resulting in an experimental setup that does not always reflect advances in modeling practices since the Sea-level Response to Ice Sheet Evolution (SeaRISE Bindschadler et al., 2013; Nowicki et al., 2013) project, including calving and frontal ablation. "

Please rephrase to better reflect the ISMIP6 standard and open protocols. The open protocol allowed groups to include calving and frontal ablation, and any other advances since SeaRISE.

We decided not to change this section. We write "does not always reflect...", which seems appropriate given that the  "open" experiments take a back seat in Goelzer 2020.

L. 142-144:  "Each group decided on the best parameter set for their simulations. This means that each model contributes a point estimate consisting of a single 'best' model run to the larger ensemble"

This statement is incorrect, since the protocol does not limit the number of simulations per group. Some groups submitted multiple runs with different physics options, resolutions, initial states, etc.

Wording changed to reflect the protocol.

L. 146: "While it is difficult to gauge the magnitude of this underestimation, Aschwanden et al. (2019) suggest that the parametric uncertainty (inter-quartile range) at 2100 is 0.3 and 12.9 cm SLE for RCP 2.6 and 8.5, respectively, which is larger than the model uncertainty suggested by the ISMIP6 experiments (0.8 and 3.4 cm SLE, respectively)."

As pointed out above, some of the uncertainties in Aschwanden et al. (2019) would need to be mapped to forcing uncertainties in ISMIP6, in order to meaningfully compare the uncertainty in the two studies.

Once again, the division between models is not entirely clear, but we argue that this distinction is immaterial with respect to the uncertainty bounds on sea level rise reported in AR6: they are either believable or they are not, regardless of where in the stack of physics uncertainty arises.

L. 148: "If one takes the Aschwanden et al. (2019) estimate of parametric uncertainty as reasonable, then the variance in ISMIP6's predictive distribution is greatly underestimated with respect to the real variance"

There are reasons to think that the Aschwanden et al. (2019) estimate of parametric uncertainty may be too wide:

(1) The range of sampled PDD factors in their simplified melt model dominates their uncertainty. That range was motivated by measurements of PDD factors in the field. However, comparison with results of similar models in GrSMBMIP (https://doi.org/10.5194/tc-14-3935-2020) shows that PDD factors that produce reasonable simulations of the historical SMB are much lower than the upper range used in Aschwanden et al. (2019). Incidentally, the way the forcing is designed in Aschwanden et al. (2019) does not allow them to constrain their PDD factors based on their own historical simulations: a problem that prevents the second premise proposed in this commentary (requiring a projection system to reproduce historical behaviour) from being applied to those results.

(2) The Aschwanden et al. (2019) projections have a shortcoming in the forcing protocol, using spatially uniform temperature forcing that likely overestimates temperatures in the ablation zone. Taken together, these points suggest that results from Aschwanden et al. (2019) may be biased in both maximum contribution and uncertainty range and are not simple to interpret in relation to other estimates.

Please provide a stronger argument as to why the Aschwanden et al. (2019) projections should be considered as reasonable, bearing in mind the above comments.

We do not make the claim that Aschwanden et al's projections are more realistic. Rather, we include this comparison as a means to illustrate the potential for parametric uncertainty to contribute non-negligible variance to predictions. Indeed, we agree that Aschwanden et al 2019 (and other papers that perform parametric ensembles) is subject to many of the same concerns that we raise for ISMIP6: it includes only one type of uncertainty, but neglects to calibrate the resulting distribution with observations. We now write

"\citet{Aschwanden2019} suggest that the parametric uncertainty for the Greenland Ice Sheet (inter-quartile range) at 2100 could be up to 0.3 and 12.9\,cm SLE for RCP 2.6 and 8.5, respectively"

We wish to reiterate here that a primary argument of this manuscript is that *both* parametric and model uncertainties need to be accounted for when constructing ensembles, and that these ensembles then need to be calibrated with observations.

L. 155: "initial ice sheet extent varied among models by up to 17%."

How well does the 17% figure characterize the variance? Please comment on the entire distribution, not just the outliers.

We removed the sentence.

ISMIP6 guidelines, as expressed in Goelzer et al. (2020), state that several metrics should be considered to evaluate a model's representation of the initial state and possible biases (see, for example, Fig. 5 and its discussion in the text). This could be mentioned here as well.

We do not believe that discussing this issue in more depth is within scope for this commentary. However, we have rearranged the text to put this in more context.

L. 183-184: "with random climate and ocean forcings developed in collaboration with their respective modelling communities (cf. Robel et al., 2019)".

Please clarify what is meant by "random climate and ocean forcings."

We have updated the language to be more clear regarding what is meant here, and added an explicit statement that we envision these forcings as an ensemble of explicit solutions from ocean and atmospheric models.

Also, it could be noted that some ice sheet models cannot run large parameter ensembles, which would reduce the number of models able to participate

Our commentary is not intended to assess the universal practicality of our suggestions. Suffice to say, however, that we understand the difficulties in running large numbers of simulations. Nonetheless, the fact that it is challenging does not also mean that it is not necessary.

L. 192-193: "To address both of these problems simultaneously, we advocate for conditioning ensemble predictions on relevant observations."

Are there other examples in the literature that could illustrate this approach?

Aschwanden, Adalgeirsdottir, and Khroulev (2013) comes to mind as a manuscript that discusses the issue, and Nias 2019 as an example for Bayesian calibration. We added both references.

L. 247: "International governments directly support development, maintenance, and operation of the Earth System Models that serve as the foundation for CMIP6 (Eyring et al., 2016), and this financial support has contributed to a suite of models that now convincingly reproduce observed climate variability (Jones et al., 2013). It is time to similarly bring ice sheet modeling to an operational level and support it with the funding the problem deserves."

Here, the authors might mention the work in progress to couple ISMs within ESMs.

Reworded.

L. 265: Data availability

Please acknowledge the use of ISMIP6 data, with text guidance available from http://www.climate-cryosphere.org/wiki/index.php?title=ISMIP6_Publication_List.

Done

L. 270: "H. Goelzer, pers. comm., November 2020"

Please update to reference Goelzer et al. (2020). The data are the same as given and displayed in the paper, e.g. in Fig. 7.

Reworded.

**References**

Mastrandrea, M.D., C.B. Field, T.F. Stocker, O. Edenhofer, K.L. Ebi, D.J. Frame, H. Held, E. Kriegler, K.J. Mach, P.R. Matschoss, G.-K. Plattner, G.W. Yohe, and F.W. Zwiers, 2010: Guidance Note for Lead Authors of the IPCC Fifth Assessment Report on Consistent Treatment of Uncertainties. Intergovernmental Panel on Climate Change (IPCC). Available at <http://www.ipcc.ch>.

Nowicki, S., Payne, A. J., Goelzer, H., Seroussi, H., Lipscomb, W. H., Abe-Ouchi, A., Agosta, C., Alexander, P., Asay-Davis, X. S., Barthel, A., Bracegirdle, T. J., Cullather, R., Felikson, D., Fettweis, X., Gregory, J., Hatterman, T., Jourdain, N. C., Kuipers Munneke, P., Larour, E., Little, C. M., Morlinghem, M., Nias, I., Shepherd, A., Simon, E., Slater, D., Smith, R., Straneo, F., Trusel, L. D., van den Broeke, M. R., and van de Wal, R.: Experimental protocol for sealevel projections from ISMIP6 standalone ice sheet models, Cryosphere, 14, 2331–2368, https://doi.org/10.5194/tc-14-2331-2020, 2020.

Slater, D. A., Straneo, F., Felikson, D., Little, C. M., Goelzer, H., Fettweis, X., and Holte, J.: Estimating Greenland tidewater glacier retreat driven by submarine melting, Cryosphere, 13, 2489-2509, https://doi.org/10.5194/tc-13-2489-2019, 2019.

Slater, D. A., Felikson, D., Straneo, F., Goelzer, H., Little, C. M., Morlighem, M., Fettweis, X., and Nowicki, S.: Twenty-first century ocean forcing of the Greenland ice sheet for modelling of sea level contribution, Cryosphere, 14, 985-1008, https://doi.org/10.5194/tc-14-985-2020, 2020.

**Nicolas Jourdain**

**Conflict of interest**:

I was involved in the working group that designed the ISMIP6 ocean forcing for Antarctica (Jourdain et al. 2020) and I am one of the numerous co-authors of several ISMIP6 papers, although not those on Greenland. I am not part of the core team that framed and lead ISMIP6.

**Summary and recommendation:**

This paper is a comment on the relevance of using the Greenland ISMIP6 ensemble as a base for anticipation and mitigation of future sea level. The authors present four types of uncertainty (model, initial state, parameters, aleatoric) and explain that ISMIP6 does not account for most of them, and therefore likely underestimates the range of uncertainty on future sea level rise. They propose a path forward that consists of running more simulations to further sample uncertainty and conditioning ensemble predictions on observations. Then, they state that a volunteer effort such as ISMIP6 is under-resourced, e.g. compared to CMIP6. This short communication can be useful for the ISMIP community, but several aspects could be improved to make it more relevant (see details below). I therefore recommend this manuscript for publication after a major revision.

We thank the reviewer for his time and insightful commentary.

**Major comments:**

1- The authors are quite critical of the ISMIP6 design, but they should acknowledge that ISMIP is very new compared to CMIP which started in 1995. It is therefore normal that ice sheet projections are not as mature as ocean–atmosphere projections. I am sure that the ISMIP community is well aware of some of the mentioned limitations and this will hopefully be improved in the future MIPs. I nonetheless agree that the range provided by ISMIP6 or its statistical emulation should not be considered as a comprehensive estimate of uncertainty. But this is also true for the CMIP6 ensemble in which the parametric uncertainty is poorly or not evaluated.

We certainly appreciate the relative immaturity of ice sheet modeling, relative to climate modeling, and state this explicitly during the close of our commentary. We also are quite confident that the ISMIP community knows of the limitations of their models. Nobody knows a model's failings better than the modelers themselves. However, the target audience of our commentary is at least as much the policy makers, funders, and broader climate science community as it is the ice sheet modeling community. And we feel that your point that ISMIP6 nor emulation products represent a comprehensive estimate of uncertainty is lost on the public that needs to know this. Indeed, the estimates and uncertainties from Edwards et al. (2021) have found themselves enshrined at the heart of the IPCC's most recent report without any such clarity. At present, we are unaware of any estimates that surpass the work of Edwards or the broader ISMIP6 in terms of its sophistication and community engagement. Nonetheless, we fear that recognition of the awesomeness of the ISMIP6/Edwards efforts has the potential to diminish the

healthy skepticism that is so often at the heart of scientific progress.  We hope that this message is now more clear in the revised text.

2- It will be challenging to make ISMIP simulations match the observational trends as long as ISMIP is forced by or coupled to CMIP models because: (1) the CMIP groups build their preindustrial control simulation by running multi-centennial or multi-millennial simulations as close as possible to a steady state (although some models still drift, e.g. Sen Gupta et al. 2013). Then, they branch off their historical simulation(s) randomly from this preindustrial run and constrain the simulation with observed anthropogenic emissions since 1850. This approach does not seem compatible with ice-sheet simulations initiated from long paleoclimate spin up. (2) Although the authors claim that natural climate variability has little effect on the observed sea-level contribution (L.208-211), low-frequency natural variability may still affect the trend values and make it difficult to compare individual CMIP-based ice-sheet projections to observed trends (only one model ensemble member should match the observational time series to consider that the model is good, not all members).

With respect to (1): We agree that uncertainty in the climate forcing for the ISMIP6 simulations has the potential to yield inaccurate predictions through no inadequacy of ice sheet models.  However, we argue that this does not absolve the ice sheet modelling community of the requirement that such uncertainty be accounted for in their method for prediction, particularly if we are to ascribe predictive power to these simulations as is done in AR6.  If models don't match simulations, it is without doubt the case that either the models are not valid for the physical system being modelled or their estimates of uncertainty are too low, regardless of the source of that uncertainty, whether that be a biased forcing or transient adjustment of the model due to a mismatch between initial conditions and forcings.

With respect to (2): We do not make the stated claim that natural climate variability has little effect on the observed sea-level contribution.  We also agree that it is possible that a mismatch between climate forcing and true climate at the decadal scale may lead to the bias between mass predictions and observations. However, we argue that the appropriate response to this observation is to increase the variance in model predictions, either by running ensembles with such variability superimposed on the forcing or making a defensible assumption about the existence of uncertainty above and beyond that which the model ensemble produces.

"only one model ensemble member should match the observational time series to consider that the model is good, not all members".  We are reticent to say whether any simulation is "good" or not.  What is clear is that the trust that we place in a given simulation should be proportional to the degree to which that simulation is statistically consistent with observations proviso a carefully considered model of observational error.

3- I am not convinced by the relevance of the "aleatoric uncertainty" (L. 110-122). First of all, the chaotic nature of the atmosphere and ocean forcing is represented in individual CMIP6 simulations (this is the aim of the multiple ensemble members provided by many CMIP groups since at least CMIP3). The resulting low frequency (interannual to decadal) natural climate variability is therefore represented in the ISMIP6 forcing, although probably underestimated due to non-eddying ocean models (Penduff et al. 2018) but this is more a CMIP6 issue than an ISMIP6 issue. If the authors have in mind shorter natural variability (extreme weather or seasonal events), then MISI is probably not a good example as the ice sheet may not be sensitive to short ice-shelf basal melt variations (e.g. Favier et al. 2019), and a better example would be hydrofracturing (e.g. Robel and Banwell 2019). In this case, however, the uncertainty is probably more on the ability of ice-sheet and firn models to represent the entire hydrofracturing process than on the atmosphere forcing.

We are also not convinced that aleatoric uncertainties are the main problem here. However, given non-linear processes such as MISI or MICI, they at least have the potential to be important, and as such we include them in our accounting of uncertainty types. Much more work is needed to quantify the hunch that such random processes are less important than other sources of uncertainty.

4- The paper is mostly about ISMIP6 Greenland projections although the title seems to point to ice sheets in general. The MISI is also quite often used as an example although it is mostly relevant for Antarctica. I think that most comments are also valid for ISMIP6 Antarctica, so I recommend balancing the paper between the model intercomparison for both ice sheets.

We added ISMIP Antarctica in our figure 1. The situation there is quite different in the sense that the stated uncertainties are much larger, due to the large variation in dynamical evolution.

5- The discussion on the lack of substantial financial support assumes that ISMIP and CMIP are two distinct entities. However, several modelling groups involved in CMIP are currently working on the coupling of ice-sheet models into Earth System Models (e.g., UKESM, CESM, IPSL-CM, EC-EARTH), with accepted projection papers for some of these groups (i.e. potential contributors as soon as CMIP7). The introduction of ice sheet models into these ESMs might change the financial aspects for these ice-sheet modelling groups. This should be mentioned.

Reworded.

6- Given that there are comments on the use of ISMIP6 results in IPCC-AR6, the final version of the report (now publicly available) should be cited and described more accurately both for the likely range they provide (section 9.4.1.3 and Tab. 9.2 of IPCC-AR6) and for their attempt to estimate high-end projections (Box 9.4).

Done.

7- There should be more discussion on how to account for the deep uncertainty related to processes that are not represented, e.g., ice-sheet feedbacks to the climate system (e.g. Sadai et al. 2020), hydrofracturing and MICI (Lai et al. 2020; DeConto et al. 2021), shear margins (Lhermitte et al. 2020).

We have cited DeConto 2021 in our discussion of model uncertainty and the MICI, as well as Sadai 2020 as examples of a hypothesized process that may deeply affect model predictions through dynamic instability.  However, discussing such processes in much more depth is outside the scope of this commentary, particularly since some dynamic processes (the MICI, for example) are still controversial in their applicability.

**Other comments:**

- The title is a bit misleading as the paper is only about Greenland.

We now include Antarctica.

- L. 24: a recent ISMIP6 reference that is more related to CMIP6 is Payne et al. (2021) https://doi.org/10.1029/2020GL091741

Changed.

- L. 30-31, about "Implicit in this approach is the assumption that the ensemble of ice sheet models perfectly spans, without bias, the range of potential sea level contribution": this is not specific to ice sheet models, it is also true for the CMIP6 ensemble, and it should be noted that a single regional atmosphere model (MAR) was used to calculate the surface mass balance and melt from the CMIP6 projections (Goezler et al. 2020).

We have removed this sentence as part of our restructuring of the introduction in response to other comments.  However, we agree that all of our criticism of ice sheet models could just as easily be leveled at other coupled earth system components.  Nonetheless, we explicitly are trying to avoid a specific discussion of the uncertainties present in ensemble forcing.

- L. 31-32 about "This ISMIP6 distribution has since been adopted as the foundation for the IPCC AR6 consensus estimate of sea level contribution from ice sheets": the AR6 (section 9.4.1.3) gives likely ranges based on the emulation of the ISMIP6 ensemble (Edwards et al. 2020), not based on the raw ISMIP6 distribution; furthermore, the IPCC corrects the estimates from Edwards et al. by adding the historical trend (see Table 9.2 of IPCC-AR6). There is also an entire box on the deep uncertainty and possible high-end projections (Box 9.4 of IPCC AR6) which is made for stakeholders with low risk tolerance.

The emulator is by design bound to the data it is fed. Its use does not correct for the failure to reproduce the historical period. We attempt to make this clear in the text.

- L. 75: instead of "poorly represented", I would write "poorly or not represented".

Changed.

- L. 96-109: it may be worth mentioning that Hill et al. (2021, https://doi.org/10.5194/tc-2021-120) and Bulthuis et al. (2019) also investigated parametric uncertainty but for Antarctic projections. A limitation is that this is often done for a limited number of parameters and that it is difficult to define an acceptable range of parameter values.

Added, but under "Accounting for all sources of uncertainty"

- L. 140-151: assuming the parametric uncertainty provided by Aschwanden et al. (2019) is a reasonable estimate for all ISMIP6 models is probably a very strong assumption. First of all, some parameters used in Aschwanden et al. (2019) are related to atmosphere and ocean forcing method that significantly differ from ISMIP6 (with completely different parameters). Then the parametric uncertainty can be highly ice-sheet model dependent. Last, I guess that the methodology used to vary parameters while keeping the model trajectory consistent with paleo-climate proxies can be a matter of debates. Having said that, I agree that the parametric uncertainty was not explored in ISMIP6, and that it should be explored in future ISMIPs. This is also true for the majority, if not all, of the CMIP projections, and some groups are now using increasing computing power to quantify this uncertainty rather than increasing model resolution.

Yes, parametric uncertainty is model-dependent. We only use Aschwanden et al as an example. We certainly do not suggest that Aschwanden et al (2019) should be used to assess parametric uncertainty in a general sense. But in the absence of another option it may at least give us a sense what to expect. We reworded the text a bit to make that clearer.

---

## Author Response (AR1)

**Response to Editor Olaf Eisen:**

L21: removed "just"

Figure 1: fixed citep command

L45: changed "predictions" to "projections"

L51: we chose "credible" based on the comment by ISMIP6 authors:

To be "accurate" implies comparison to observation. Projections are of the future, so it is impossible to assess their accuracy now. We suggest a different phrasing.

However, after some reflection, we agree with the editor that "accurate" is more appropriate than "credible". It is true that the accuracy of projections cannot be assessed now, but we do not see the "assessing now" as a requirement. Changed back to "accurate". Also changed line 59 back to "accurate" for consistency.

L 62: "here it is really important to consider the rate of change. In your response you stated:
"In addition, we would like to point out that from a societal impact point of view it is the cumulative mass loss number that is of relevance, not the rate."
So this is not consistent, as especially for coastal planning the SLR rate is very important. I would therefore suggest you extend the sentence:
"preparing for future sea-level rise and rates"

-> Fair point. When writing our response to the question why using cumulative changes rather than rates, we may have not thought our response through enough. Rates do matter. However we would prefer not to change the sentence structure as "preparing for future sea level rise" is a general statement that does not exclude the rates.

L 210: changed "predictions" -> "projections"

L 287: we stick with "querying".

L 342: removed as suggested.

Figure 1 and 2: the 2-sigma of the observations are hard to see because the variance of the simulations is so large. We've tried to improve the readability of the figures by changing from green to blue (while sticking with a single hue colorscheme from colorbrewer2.org), and by changing the vertical extent.

Supplement: This figure can be found at the end of the main manuscript under Data Availability.